# Pericyte-to-Endothelial Cell Communication via Tunneling Nanotubes Is Disrupted by a Diol of Docosahexaenoic Acid

**DOI:** 10.3390/cells13171429

**Published:** 2024-08-26

**Authors:** Sebastian Kempf, Rüdiger Popp, Zumer Naeem, Timo Frömel, Ilka Wittig, Stephan Klatt, Ingrid Fleming

**Affiliations:** 1Centre for Molecular Medicine, Institute for Vascular Signalling, Goethe University, 60596 Frankfurt am Main, Germany; kempf@vrc.uni-frankfurt.de (S.K.); popp@vrc.uni-frankfurt.de (R.P.); naeem@vrc.uni-frankfurt.de (Z.N.); froemel@vrc.uni-frankfurt.de (T.F.); klatt@vrc.uni-frankfurt.de (S.K.); 2Institute for Cardiovascular Physiology, Goethe University, 60596 Frankfurt am Main, Germany; wittig@med.uni-frankfurt.de; 3German Center of Cardiovascular Research (DZHK), Partner Site RheinMain, 60596 Frankfurt am Main, Germany

**Keywords:** polyunsaturated fatty acid, dihydroxydocosapentaenoic, diabetes, pericyte-to-endothelial cell communication

## Abstract

The pericyte coverage of microvessels is altered in metabolic diseases, but the mechanisms regulating pericyte–endothelial cell communication remain unclear. This study investigated the formation and function of pericyte tunneling nanotubes (TNTs) and their impact on endothelial cell metabolism. TNTs were analyzed in vitro in retinas and co-cultures of pericytes and endothelial cells. Using mass spectrometry, the influence of pericytes on endothelial cell metabolism was examined. TNTs were present in the murine retina, and although diabetes was associated with a decrease in pericyte coverage, TNTs were longer. In vitro, pericytes formed TNTs in the presence of PDGF, extending toward endothelial cells and facilitating mitochondrial transport from pericytes to endothelial cells. In experiments with mitochondria-depleted endothelial cells displaying defective TCA cycle metabolism, pericytes restored the mitochondrial network and metabolism. 19,20-Dihydroxydocosapentaenoic acid (19,20-DHDP), known to disrupt pericyte–endothelial cell junctions, prevented TNT formation and metabolic rescue in mitochondria-depleted endothelial cells. 19,20-DHDP also caused significant changes in the protein composition of pericyte-endothelial cell junctions and involved pathways related to phosphatidylinositol 3-kinase, PDGF receptor, and RhoA signaling. Pericyte TNTs contact endothelial cells and support mitochondrial transfer, influencing metabolism. This protective mechanism is disrupted by 19,20-DHDP, a fatty acid mediator linked to diabetic retinopathy.

## 1. Introduction

The microcirculation plays a unique role in tissue homeostasis and has the capacity to adjust the supply of blood within tissues to meet the demands of specific cell populations for oxygen and nutrients. Exchange takes place across the membranes of endothelial cells that make up the capillaries, and although these cells can release vasoactive factors to promote dilation, i.e., nitric oxide or endothelium-derived hyperpolarizing factors, an intimate relation with pericytes in precapillary arterioles and postcapillary collecting venules is required for optimal function and the maintenance of barrier function. The term pericyte is now known to reflect a number of mural cell subsets and has been attributed roles in promoting and maintaining endothelial cell quiescence in newly generated vessels as well as contraction to regulate micro-environmental blood flow (for a review, see Wu et al. [1]).

The role of endothelial dysfunction in the development of cardiovascular disease has been well documented [2,3], but the contribution of alterations in pericyte function to cardiovascular and pulmonary disease development is underappreciated. Indeed, it now seems that the pericyte loss or migration away from the vasculature, which was previously described in retinas from animals and humans with diabetes [4], also occurs during the development of the microcirculatory dysfunction that accompanies heart failure [5,6]. While the classical contact between pericytes and endothelial cells seems to be via their shared basal lamina, long, fine structures that enable the transport of material over many tens of microns have been reported, which connect pericytes with other pericytes or with endothelial cells in an adjacent vessel. These structures or tunneling nanotubes (TNTs) were initially described in 2004 as “nanotubular highways” [7,8] and are now known to support the intercellular trafficking of many types of cellular compounds ranging from Ca^2+^ and electrical signals [9], micro-RNAs [10], and proteins [11,12] to preformed MHC complexes [13]. They can even transport endosomal vesicles [14], organelles such as lysosomes, and mitochondria in a process referred to as “organellar diakinesis” [15,16]. Depending on the conditions, the transfer of mitochondria through TNTs has been proposed to rescue impaired mitochondrial function in recipient cells and to propagate degenerative states linked to mutated mitochondrial DNA [17,18]. We previously linked the pericyte depletion and the loss of retinal barrier function in diabetic retina with the generation of a diol of docosahexaenoic acid (DHA), i.e., 19,20-dihydroxydocosapentaenoic acid (19,20-DHDP) [19]. Therefore, the aim of this study was to identify pericyte TNTs in vivo and in vitro to study their generation, the impact of pericyte TNTs on endothelial cell metabolism, and the role of 19,20-DHDP in these processes.

## 2. Materials and Methods

### 2.1. Animals

Ins2Akita (C57BL/6-Ins2Akita/J) mice carrying a mutation in the insulin 2 gene [20] were obtained from Jackson Laboratory. The animals were housed in conditions that conform to the Guide for the Care and Use of Laboratory Animals published by the U.S. National Institutes of Health (NIH publication no. 85–23). Mice were sacrificed using 4% isoflurane and cervical dislocation for the isolation of the retina. Only 10-month-old male mice were used in the current study.

### 2.2. Cell Culture

Pericytes: Human brain vascular pericytes were purchased from ScienCell Research Laboratories (Carlsbad, CA, USA) and cultured in DMEM/F-12 Media (11320033, Gibco, Waltham, MA, USA) supplemented with 5% FCS (Gibco, Waltham, MA, USA), insulin (Sigma-Aldrich, Taufkirchen, Germany), and human-FGF. For this study, six independent batches of pericytes were used up to passage seven.

Endothelial cells: Human umbilical vein endothelial cells were isolated and cultured, as described previously [21], in ECGM2 Media (C-22111, Promocell, Heidelberg, Germany) supplemented with 5% FCS (Gibco, Waltham, MA, USA) and the commercially available supplement mix (C-39216, Promocell, Heidelberg, Germany). The cultivated cells were used up to passage four. 

### 2.3. Endothelial Cell Pericyte Co-Culture

2D Endothelial cell pericyte co-culture: Endothelial cells were seeded on dishes coated with 1% gelatin in serum and growth factor-depleted medium containing fatty acid BSA (0.1%). Once the endothelial cells adhered, pericytes were added at a ratio of 1:3 and the mixture was co-cultured for 6 or 16 h in the presence of PDGF-BB (0.3–3 ng/mL, Hz-1308, Proteintech, Planegg-Martinsried, Germany). Experiments were performed in the absence and presence of 19,20-DHDP (10 µmol/L, Cayman, Ann Arbor, MI, USA) and in the presence of an sEH inhibitor (*t*-AUCB, 10 µmol/L). In some experiments, endothelial cells were isolated from the co-cultures at the end of the incubation period using CD144-coated magnetic beads (Miltenyi Biotec, Bergisch Gladbach, Germany). Pericytes were collected from the flowthrough and were enriched by centrifugation. Thus, isolated cells were directly processed for immunophenotyping, and for the rest of the experiments, the cells were, after separation, frozen in liquid nitrogen and further processed as described.

3D Filter-based co-culture: Endothelial cells were seeded at a high density (600,000 cells per well) on the underside of gelatin-coated six-well Transwell filter inserts (0.4 µm Sarstedt, Nümbrech, Germany). After 6 h, the inserts were inverted and pericytes (600,000 per well) were seeded on the opposite side of the filter. The co-culture was maintained for 5 days to ensure that endothelial cells and pericytes were confluent and were able to form endothelial cell–pericyte junctions in the pores of the filter, as described [19]. The co-cultures were then treated with either the solvent (0.1% DMSO) or 19,20-DHPD (10 µmol/L) in the presence of the sEH inhibitor (*t*-AUCB, 10 µmol/L) for 12 h. The cells were removed by scraping both sides of the filter, and proteins in the filter pores were isolated by incubating the filter with RIPA-lysis buffer (Tris/HCL 50 mmol/L pH 7.5, NaCl 150 mmol/L, NaPi 10 mmol/L, NaF 20 mmol/L, 1% sodium deoxycholate, 1% Triton, and 0.1% SDS) and recovered by centrifugation (15,000 rpm, 15 min, 4 °C). 

### 2.4. TNT Formation 

To study TNT formation, pericytes (10,000) were seeded on a 1% gelatine-coated µ-slide 8 well (ibidi, Gräfelfing, Germany), and when ~40% confluent, the medium was replaced with serum and growth factor-depleted medium containing fatty acid free bovine serum albumin (BSA; 0.1%) and either the solvent (0.01% DMSO) or PDGF-BB (0.3–3 ng/mL) for 6 h. The experiments were performed in the absence and presence of 19,20-DHDP (10 µmol/L) and stopped by the addition of 4% paraformaldehyde (PFA). TNTs were visualized using a confocal microscope (Carl Zeiss LSM 780, Jena, Germany) by rendering the average phalloidin-546 signal and quantifying the length in Fiji ImageJ (version 1.53t, NIH).

### 2.5. Detection of Mitochondria and Mitochondrial Exchange

Mitochondria were visualized after incubating cells with MitoTracker Red (1:1000, 22426, Thermo Fisher, Waltham, MA, USA) or MitoTracker Green (1:1000, M7514, Thermo Fisher) for 2 h. Thereafter, the cells were washed extensively with phosphate buffered saline (PBS) and cultured as described above. The MitoTracker signals were visualized in cells treated with either the solvent (0.1% DMSO), PDGF-BB (0.3–3 ng/mL) or 19,20-DHDP (10 µmol/L) for 6 h, and imaging was performed using a confocal laser scanning microscope (LSM 780, Zeiss, Jena, Germany). In some experiments, the MitoTracker Red signal was assessed by flow cytometry (BD LSRFortessa X-20, BD, Heidelberg, Germany) in CD31+ endothelial cells isolated from 2D endothelial cell-pericyte co-cultures. The data were analyzed using FlowJo Vx software (version 18.8.1, TreeStar, Ashland, OR, USA).

### 2.6. Mitochondria Depletion

The selective deletion of mitochondrial DNA (mtDNA) was induced as described [22,23] by culturing freshly isolated endothelial cells in endothelial cell growth medium (ECGM)-2 (PromoCell, Heidelberg, Germany) containing ethidium bromide (25 µmol/L; Roth, Karlsruhe, Germany) and uridine (50 µg/mL; Sigma-Aldrich, Taufkirchen, Germany) for 14 days. Mitochondria-depleted cells were selected using carbonyl cyanide m-chlorophenyl hydrazone (CCCP, 1 ng/mL; C2920, Sigma-Aldrich, Taufkirchen, Germany).

### 2.7. Immunohistochemistry

Isolated retinas and cells were fixed in paraformaldehyde (PFA, 4%) for 2 h at room temperature. The samples were then blocked and permeabilized in PBS containing 0.5% Triton X-100 and 5% donkey serum (4 h, room temperature). After permeabilization, the samples were incubated with primary antibodies overnight (4 °C), and after washing, they were incubated with appropriate secondary antibodies (1:200 in PBS) together with DAPI (10 ng/mL, ThermoFisher, Waltham, MA, USA) for one hour. The samples were then washed and mounted in Fluoromount-G (ThermoFisher, Waltham, MA, USA). 

The following primary antibodies were used: NG-2 (1:200, AB5320, Millipore, Darmstadt, Germany), CD31 (1:100, 557355, BD Biosciences, Heidelberg, Germany), connexin 43 (1:100, 3512, Cell Signaling, Leiden, Netherlands), VDAC (1:100, sc390996, Santa Cruz, Heidelberg, Germany), Tom20 (1:100, sc17764, Santa Cruz, Heidelberg, Germany). The Alexa Fluor-coupled secondary antibodies were from Thermo-Fisher (1:200, Waltham, MA, USA) or fluorophore conjugated phalloidin-546 (1:1000, A22283, Thermo Fisher, Waltham, MA, USA).

### 2.8. Immunoblotting

For immunoblotting, the cells were lysed in RIPA lysis buffer (Tris/HCL 50 mmol/L, pH 7.5, NaCl 150 mmol/L, NaPi 10 mmol/L, NaF 20 mmol/L, 1% sodium deoxycholate, 1% Triton X-100, and 0.1% SDS containing protease and phosphatase inhibitors), and detergent-soluble proteins were suspended in SDS-PAGE sample buffer. The samples were separated by SDS-PAGE and subjected to Western blotting, as described [24]. Membranes were blocked in Rotiblock (Roth, Karlsruhe, Germany) for 2 h and incubated overnight at 4 °C with primary antibodies. Following extensive washing, horseradish peroxidase-conjugated secondary antibodies were added for 2 h, and protein bands were visualized by using Lumi-Light plus (Roche, Mannheim, Germany). Luminescence was captured using an image acquisition system (Fusion FX7; Vilber-Lourmat, Torcy, France). 

The following primary antibodies were used: anti-phospho Tyr751 PDGFR (1:1000, Cat. # 3161), anti-PDGFR (1:1000, Cat. # 3169), anti-phospho Ser473 Akt (1:1000, Cat. # 4058), anti-Akt (1:1000, Cat. # 9272), and anti-phospho Thr202/Tyr204 Erk1/2 (1:1000, Cat. # 9106), which were all from Cell Signaling (Leiden, The Netherlands). The antibody against Erk1/2 (1:1000, Cat. # ABS44) was from Sigma-Aldrich (Taufkirchen, Germany), the antibodies recognizing VDAC (1:1000, Cat. # sc390996) and Tom20 (1:1000, Cat. # sc17764) were from Santa Cruz (Heidelberg, Germany), and the anti β-actin (1:3000, Cat. # MAK6019) was from Linaris (Eching, Germany). The peroxidase conjugated secondary antibodies used at a dilution of 1:20,000 in Tris-buffered-saline-Tween were goat anti-rabbit (Cat #. 401393) and goat anti-mouse (Cat. # 401253), both from Merck (Darmstadt, Germany).

### 2.9. Measurement of Mitochondrial Oxidation by MitoSOX

Mitochondrial ROS production was assayed as described [25], with some modifications. Mitochondrial depleted endothelial cells were cultured with and without pericytes in the presence of PDGF-BB (1 ng/mL) or PDGF-BB and 19,20-DHDP (10 µmol/L) for 12 h. Thereafter, the cells were incubated with MitoSOX (1 µmol/L; ThermoFischer) for 30 min at 37 °C. The cells were recovered by enzymatic digestion and incubated with antibodies directed against CD31 (1:100, Cat #. 561653, BD-Biosciences, Heidelberg, Germany). The MitoSOX and MitoTracker Red signals in CD31+ endothelial cells were measured by flow cytometry (BD LSRFortessa X-20, BD, Heidelberg, Germany), and the data were analyzed using FlowJo Vx software (version 18.8.1, TreeStar, Ashland, OR, USA).

### 2.10. Proteomics

The mass spectrometry proteomics data including a detailed method description and data analysis have been deposited to the ProteomeXchange Consortium via the PRIDE [26] partner repository, with the dataset identifier PXD049131. 

MaxQuant Search data were further analyzed using Perseus 1.6.1.3 [27]. Reverse identifications and common contaminants were removed. Proteins were filtered to be identified at least three times (n = 3) in one experimental group. Missing values were replaced by the lowest value in the matrix. Most affected proteins were determined by Student’s *t* test and permutation-based FDR (q value). 

### 2.11. Metabolomics

Sample extraction: Trifluoroethanol:H_2_O (1:1) was added to each sample. The samples were shortly vortexed and incubated on ice for 10 min. Next, methanol:ethanol (1:1) was added and the samples were incubated for another 10 min. Ice-cold water, containing internal standards—13C-fumarate-C2, 13C-citrate-C2 and 13C-succinate-C1,4 (all 0.02 mmol/L)—was added. After 10 min, the samples were sonicated on ice for 30 s and centrifuged (14,000 rpm, 10 min, 4 °C). The supernatant was recovered and transferred to two separate tubes before being evaporated under a stream of N2 gas. The remaining aqueous fraction was shock-frozen in liquid nitrogen and subsequently freeze-dried (Alpha 3–4 LSCbasic system, Martin Christ, Osterode am Harz, Germany). 

The dried samples were reconstituted in H_2_O:acetonitrile (AcN; 1:3) for the analysis of pyruvate, 2-HG, succinate, malate, and fumarate (method 1—HILIC separation) or in 100% H_2_O containing 0.2% formic acid for the analysis of the remaining metabolites of the TCA cycle and the amines asparagine, glutamine, and glutamate (method 2—C18 separation). The samples were centrifuged for 10 min at 14,000 rpm and 4 °C and the supernatant was transferred to MS vials and subjected to mass spectrometric analysis.

Quantification and analysis: Method 1 (HILIC separation): The dried samples were reconstituted in H2O: AcN (1:3) and then centrifuged (14,000 rpm, 4 °C, 10 min), and the supernatant was transferred to mass spectrometry (MS) vials. Thereafter, the samples (4 µL) were injected via an Infinity II Bio liquid chromatography system into a 6495C triple quadrupole mass spectrometer (both Agilent Technologies, Waldbronn, Germany). The metabolites were separated on an Acquity BEH Amide HILIC column (1.7 µm, 2.1 × 100 mm, Waters) using the following mobile phase binary solvent system and gradient at a flow rate of 0.4 mL/min. Mobile phase A consisted of 100% water with 10 mmol/L ammonium acetate and 5 µmol/L medronic acid (pH 9.3). Mobile phase B consisted of 100% acetonitrile with 10 mmol/L ammonium acetate. The following 23 min gradient program was used: starting with 85% B, 0–1 min 85% B, 1–8 min 75% B, 8–12 min 60% B, 12–15 min 10% B, 15–18 min 10% B, 18–19 min 85% B, and 19–23 min 85% B. The column compartment was set to 20 °C. Metabolites were detected with authentic standards and/or via their accurate mass, fragmentation pattern, and retention time in polarity switching ionization dynamic MRM AJS-ESI mode and quantified (where appropriate) via a calibration curve. The gas temperature of the mass spectrometer was set to 200 °C, the gas flow was 14 L/min, the nebulizer was set to 40 psi, and the sheath gas flow was 12 L/min, 375 °C. The capillary voltages were set at 4000/2500 V, with a nozzle voltage of 500/0 V. The voltages of the High-Pressure RF and Low-Pressure RF were set to 150/90 and 60/60 V, respectively. The metabolite peaks were annotated with Skyline-daily (version 22.2.1.278). Quantification was achieved by comparing sample values with the slope of a standard curve of the appropriate standard, and the data are presented as the concentration per µg protein.

Method 2 (C18 separation): The dried samples were reconstituted in H_2_O containing 0.2% formic acid and centrifuged (14,000 rpm, 4 °C, 10 min), and the supernatant was transferred to MS vials. Thereafter, the samples (8 µL) were injected as described above. The metabolites were separated on an Acquity HSS T3 column (1.8 µm, 2.1 × 150 mm, Waters, Eschborn, Germany) using the following mobile phase binary solvent system and gradient at a flow rate of 0.35 mL/min: mobile phase A consisted of H_2_O containing 0.2% formic acid, and mobile phase B consisted of acetonitrile containing 0.2% formic acid. The following 11 min gradient program was used: starting with 1% B, 0–6 min 1% B, 6–7 min 80% B, 7–8 min 80% B, and 8–11 min 1% B. The column compartment was set to 30 °C. Metabolites were detected as mentioned above. The gas temperature was 240 °C, the gas flow 19 L/min, the nebulizer was set to 50 psi, and the sheath gas flow was set to 11 L/min (400 °C). The capillary voltages were set at 1000/1000 V, with a nozzle voltage of 500/500 V. The voltages of the High-Pressure RF and Low-Pressure RF were set to 100/100 and 70/70 V, respectively. Metabolite peaks were annotated with Skyline-daily (version 22.2.1.278). Quantification was achieved by comparing the sample values with the slope of a standard curve of the appropriate standard, and the data are presented as the concentration per µg protein.

### 2.12. Statistical Analyses

The data are expressed as the mean ± SEM. Statistical evaluation was performed using Student’s *t* test for unpaired data. One-way ANOVA (Bonferroni) and two-way ANOVA (followed by Tukey’s multiple comparisons test) were used, where appropriate. The statistics for metabolomics were performed with Metaboanalyst 5.0. The statistical tests are described in the figure legend for each experiment. Values of *p* < 0.05 were considered statistically significant.

## 3. Results

### 3.1. Tunneling Nanotubes and Migrating Pericytes in the Retina

The pericyte coverage of vessels in the second capillary layer of retinas from 10-month-old wild-type mice was high (approximately 50%), and some cells were observed to extend long processes (7–30 µm) that spanned capillaries (Figure 1a). Such processes have been classified as TNTs [7,8,28], and although, previous reports have suggested that 30% of retinal pericytes extrude them [29]; the incidence we observed was much lower. In retinas from diabetic Ins2Akita mice, pericyte coverage was decreased, i.e., approximately 20% (Figure 1b), which is consistent with the reported pericyte migration and loss in this model [4,19]. Despite this, more TNTs were apparent in retinas from diabetic mice than in non-diabetic mice (Figure 1c), and they were significantly longer with the size distribution skewed to higher values (Figure 1d). Three distinct pericyte morphologies were distinguishable in retinas from diabetic mice: (1) pericytes that were associated closely with endothelial cells and did not form inter-capillary bridges, (2) pericytes whose nuclei were aligned with capillaries and extruded long processes (7–70 µm), and (3) a subpopulation with nuclei that were clearly no longer aligned with endothelial cells and, thus, defined as migrating pericytes (Figure 1e). 

### 3.2. Tunneling Nanotubes Transport Mitochondria between Pericytes and Endothelial Cells

Pericytes in culture spontaneously formed TNTs/TNT-like structures that were rich in F-actin (Figure 2a). The number and length of these structures increased in the presence of different concentrations of PDGF BB (Figure 2b). Consistent with a previous report [30], connexin 43 containing gap junctional plaques were observed at the contact points between pericyte TNTs and endothelial cells, as well as within TNTs (Figure 2c).

TNTs support the intercellular trafficking of many cellular compounds and organelles; such as mitochondria [15,16,31,32,33]. As pericytes are usually in close contact with endothelial cells, we determined whether or not mitochondria could be exchanged between these two cell types. To this end, endothelial cells containing MitoTracker Red and pericytes containing MitoTracker Green were seeded in separate inserts on one slide and allowed to form confluent monolayers (Figure 3a). Thereafter, the inserts were removed, and the cells were allowed to migrate towards each other in the presence of PDGF BB. The first contact detected between the two cell types was in the form of MitoTracker Red-containing pericyte TNTs extending towards endothelial cells (Figure 3b). The fact that mitochondria were contained in the growing TNTs was confirmed by the staining of the mitochondrial protein and the voltage-dependent anion-selective channel (VDAC) (Figure 3c). When contact was established between pericytes and endothelial cells, mitochondrial transfer took place, as evidenced by the appearance of endothelial cells containing both MitoTracker Green and MitoTracker Red (Figure 3d). This unidirectional transport, i.e., from pericytes to endothelial cells, was also observed when the mitochondrial dyes were reversed (Appendix A). We did not detect any evidence of mitochondrial transfer from endothelial cells to pericytes, an observation that fits with reports indicating that organelle transport is frequently unidirectional, i.e., from the cell that initiated the formation of TNT to the receptor cell [8,34].

### 3.3. Impact of Pericyte-Derived Mitochondria on EtBr-Treated Endothelial Cells

To study the consequences of mitochondrial transfer between the two cell types in more detail, experiments were performed using human endothelial cells cultured in the presence of ethidium bromide and uridine (EtBr) for 14 days to prevent mitochondrial DNA (mtDNA) replication and thus deplete cells of mitochondria [23]. The approach used disrupted the mitochondrial network but did not result in the complete loss of the mitochondrial protein Translocase of outer mitochondrial membrane 20 (Tom20) (Figure 4a) or the MitoTracker Red signal (Figure 4b). However, the expression of Tom20 and VDAC was markedly reduced (Figure 4c) and accompanied by an increased generation of mitochondrial reactive oxygen species (ROS) (Figure 4d). While the latter effect seemed counterintuitive, the EtBr-induced decrease in mitochondrial DNA in fibroblasts was also associated with increased ROS generation [35], which could be attributed to residual mitochondria being dysfunctional. 

Next, EtBr was removed and endothelial cells were co-cultured with pericytes for 6 h. This resulted in mitochondria-filled TNTs emanating from pericytes towards endothelial cells to enable the reformation of a mitochondrial network that was labelled with MitoTracker Green and pericyte-derived MitoTracker Red (Figure 5a). Consistent with observations made using the marker dyes, pericyte-derived mitochondria restored the protein levels of Tom20 and Translocase of inner mitochondrial membrane 17 (Tim17) in endothelial cells isolated from the co-culture using CD144-coated magnetic beads (Figure 5b). The expression of VDAC in endothelial cells was partially recovered during the 6 h of EtBr washout, but the co-culture with pericytes markedly increased its levels, clearly indicating the impact of mitochondrial transfer from pericytes. 

We previously reported that an increase in the expression of the sEH in the retina and the increased generation of the DHA diol, 19,20-dihydroxydocosapentaenoic acid (19,20-DHDP), were responsible for the dissolution of pericyte endothelial cell junctions and the development of non-proliferative diabetic retinopathy [19]. Therefore, we determined the effect of 19,20-DHDP on the formation of pericyte TNTs in vitro. While TNTs rapidly developed in PDGF-treated pericytes, the TNTs that formed in the presence of 19,20-DHDP were clearly shorter and formed fewer contacts with other pericytes (Figure 5c). Treating pericytes with the sEH substrate 19,20-EDP (in the presence of an sEH inhibitor to prevent any metabolism to 19,20-DHDP) had a slight effect on the TNT length but no effect on the number of TNTs over 100 µm. Moreover, while pericytes rescued mitochondria numbers and free radical production in CD31+ endothelial cells, the effects were blunted when pericytes were treated with 19,20-DHDP (Figure 5d–f).

### 3.4. 19,20-DHDP Antagonizes the Rescue of Endothelial Cell Metabolism by Pericytes

There was a clear impact of endothelial cell mitochondrial depletion on the initial part of the TCA cycle, i.e., markedly reduced intracellular levels of citrate and isocitrate (Figure 6). Cells depleted of mtDNA with EtBr are known to maintain some mitochondrial function [36], and we observed that levels of α-ketoglutarate (α-KG), succinate, and fumarate were elevated in EtBr-treated cells, indicating that residual endothelial cell mitochondria were able to partly refuel the TCA cycle using glutamine as an energy source. Indeed, the glutamine levels were unexpectedly high, which likely reflects an increase in its uptake. α-KG was also metabolized to 2-hydroxyglutarate (2HG), the levels of which were also markedly elevated in EtBr-treated endothelial cells. To determine the impact of pericyte-derived mitochondria on endothelial cell metabolism, EtBr-treated cells were washed and incubated with PDGF-treated pericytes overnight. Thereafter, endothelial cells were separated by CD144-coated magnetic beads and subjected to metabolomic analyses via LC-MS/MS. The co-culture with pericytes restored the TCA cycle metabolite levels to those detected in mitochondria-intact (solvent-treated) endothelial cells. Pericytes also decreased the endothelial cell levels of glutamine but had a minor impact on α-KG and 2HG, which can be accounted for by the re-establishment of the TCA cycle. Consistent with our observations on TNT formation, the pericyte-associated rescue of the endothelial cell TCA cycle was largely prevented in cells exposed to 19,20-DHDP. 

### 3.5. Effect of 19,20-DHDP on Pericyte-Endothelial Cell Junction Proteins

To gain insight into the impact of 19,20-DHDP on intracellular communication, endothelial cells and pericytes were placed on opposite sides of a filter insert and cultured to confluence. Thereafter, the filters were treated with the solvent or 19,20-DHDP before the endothelial cells and pericytes were carefully removed and the proteins present in the filter pores were analyzed by mass spectrometry. The endothelial cell–pericyte junctions contained a large number of proteins (Figure 7a,b). Overall, 1003 proteins were detected in junctions in the presence of the solvent, and a discrete population of those were depleted from junctions treated with 19,20-DHDP and included transport proteins such as the cationic amino acid transporter, SLC7A5, and the alanine-serine-cysteine transporter, SLC1A5. Pathway analysis of the junctional proteins altered by 19,20-DHDP identified phosphatidylinositol 3-kinase, PDGF receptor β, and RhoA-related signaling events as being altered by the DHA diol (Figure 7c). As PDGF is essential for endothelial cell–pericyte communication and coverage [37,38] and stimulated TNT formation, we determined whether or not 19,20-DHDP was able to interfere directly with PDGF signaling in pericytes. This seemed a logical hypothesis given that this lipid mediator can alter the cholesterol content and rearrange protein complexes in lipid rafts that may well affect receptor signaling [39]. However, 19,20-DHDP did not directly affect the ability of PDGF to elicit the phosphorylation of its receptor (PDGFR) or the subsequent phosphorylation of either AKT or ERK1/2 (Figure 7d). Thus, although the fatty acid diol prevented PDGF-induced TNT formation and altered the expression of proteins involved in PDGF signaling, it had no direct impact on the early activation of the PDGF receptor.

## 4. Discussion

The results of the present investigation highlight that a metabolic communication exists between pericytes and endothelial cells, which involves the transfer of functional mitochondria through TNTs. Moreover, a DHA diol, i.e., 19,20-DHDP, generated by sEH, which promotes the dissolution of endothelial cell–pericyte junctions to initiate pericyte migration in vivo [19], stunted the formation of TNT-like structures and prevented the metabolic rescue of mitochondrial-depleted endothelial cells by pericytes.

Pericytes are key regulators of neurovascular coupling and have been proposed to contract to regulate blood flow (for reviews, see references [40,41]). It has also been suggested that pericytes act as metabolic sentinels through ATP-sensitive K^+^ channels, opening when ATP levels drop and eliciting pericyte hyperpolarization. The latter was proposed to be transferred to endothelial cells to elicit an ascending dilation, i.e., the relaxation of an upstream arteriole to increase blood flow and restore glucose delivery [42]. The thin inter-pericyte TNTs that have been characterized to date may also play a role in the coordination of blood flow, as they could act as electrical signaling shortcuts, particularly in organs in which TNTs connect two distinct capillaries [29]. To date, TNTs have been roughly classified into two groups: those with a diameter of less than 0.7 µm, which are primarily composed of actin and carry small membranous cargoes, and those with a diameter larger than 0.7 µm, which contain both actin and microtubules and are capable of transporting larger cytoplasmic components (for a review, see Dagar and Subramaniam [43]). Other authors refer to a class of pericyte extension that terminates on other cell types as tunneling microtubes [41]. These seem to be more common in the heart than in the retina and can run along a capillary for tens of microns or more [41]. Just as the exact role played by TNTs in the regulation of microcirculatory function is not known, it is equally unclear whether and how the consequences of their loss could be linked with microcirculatory dysfunction. Examples of the latter state could be glaucoma, where inter-pericyte TNTs are damaged by ocular hypertension [44], or even heart failure with preserved ejection fraction, where pericyte loss seems to precede microvascular rarefication [5,6]. The latter situation is highly reminiscent of non-proliferative diabetic retinopathy, where pericytes become detached from, and migrate away from, capillary endothelial cells [4]. During this process, any TNT-mediated metabolic surveillance or coordinating role in blood flow would be lost, a situation that could affect blood flow distribution and contribute to disease progression. 

What stimulates the formation of TNTs? At least in cultured pericytes, TNT formation was limited unless PDGF was added to the culture medium. In the microcirculation, this factor could be secreted by target cells such as endothelial cells and thus may act as a TNT-inducing signal, e.g., in response to hypoxia [45]. In other cell types, the process may well be a response to an inflammatory insult, as tumor necrosis factor-α can also initiate TNT formation [46], as can viral infection [47]. The durability of these structures is also not clear, as TNTs can be transient, and reports of their existence vary from minutes to several hours [43]. The latter would be expected to be required for metabolic regulation and the intercellular transfer of organelles. While it is highly likely that even more sub-types of TNT that are regulated by distinct control mechanisms exist, some of the observations made may reflect the importance of different molecules in the docking and maturation of TNTs. Indeed, even though the pericyte TNTs co-cultured with endothelial cells were able to transfer mitochondria, we detected high levels of connexin 43, which would fit better with electrical communication, when initial contact was made. Pericytes seem to be more active in generating TNTs than endothelial cells [28], and the former actively explore their microenvironment for appropriate sites to dock onto adjacent pericytes or endothelial cells. Endothelial cells, on the other hand, may well communicate over long distances via secretory granules and extracellular vesicles [48]. This fits well with our observation that PDGF stimulates pericyte TNT formation and the fact that this growth factor is secreted by endothelial cells [37].

As no TNT signature protein has been identified yet, we determined the protein composition of the endothelial cell–pericytes junctions formed in a filter co-culture model. A number of proteins were detected within endothelial cell–pericyte junctions that were no longer detected in cells treated with the sEH product, 19,20-DHDP. A smaller number of proteins increased in junctions from cells treated with the diol. Of the 50 targets that were most affected by 19,20-DHDP, there were proteins involved in PDGFRβ signaling, cellular trafficking, and the mitochondria. Among the latter were RhoA and Rho GDP dissociation inhibitor α (ARHGDIA), which is of relevance inasmuch as a Rho kinase inhibitor was reported to improve the efficiency of intercellular mitochondrial transport between human corneal epithelial cells [49]. How could 19,20-DHDP affect the growth and targeting of TNTs? A series of fundamental steps are essential for the formation of functional nanotubes, including a controlled and directed F-actin polymerization for outgrowth and elongation, local guidance cues to direct the nascent TNT to its target, and the activation of processes to facilitate membrane fusion to create a functional transport route (for a review, see Dagar et al. [50]). It has been suggested that the lipid bilayer composition might favor membrane fusion between the TNT tip and the targeted cell [51,52]. Indeed, cholesterol-sphingomyelin nanodomains lipids have been detected in TNTs generated by a cancer cell line, and cholesterol depletion reduced the number of TNTs generated [53]. Similar observations were made using mast cells [54]. Given that 19,20-DHDP interferes with the lipid composition of the membrane to impact cell function and cell–cell communication, it is perhaps not surprising that this lipid mediator interferes with TNT docking to target cells. Little is known about the actions of 19,20-DHDP, but like its parent PUFA, i.e., DHA, it is able to alter membrane dynamics and displace cholesterol as well as cholesterol-linked proteins from lipid rafts [19,39,55]. From the data presented, it also seems that 19,20-DHDP interferes with PDGF B signaling, as it prevented the PDGF-induced formation of TNTs. However, despite the fact that proteins involved in PDGF singling were depleted from endothelial–pericyte junctions in the presence of 19,20-DHDP, the diol did not affect the initiation of PDGF receptor signaling, i.e., the phosphorylation of the receptor.

## Figures and Tables

**Figure 1 cells-13-01429-f001:**
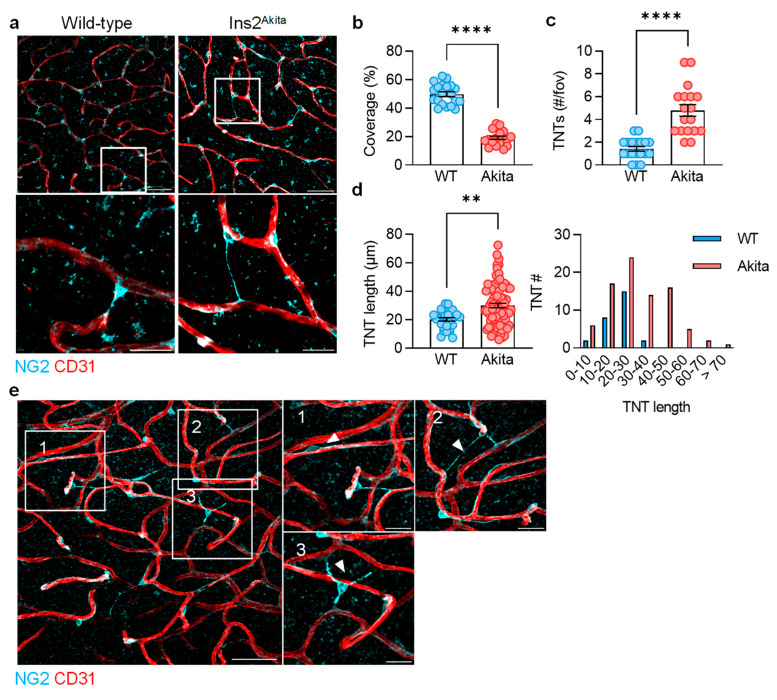
Pericyte tunneling nanotubes (TNTs) in retinas from non-diabetic and diabetic mice. Retinas were obtained from diabetic 10-month-old Ins2^Akita^ mice and their non-diabetic littermates. (**a**) Pericytes (NG2) and capillaries (CD31) in the second capillary layer of retinas from non-diabetic and diabetic Ins2^Akita^ littermates; upper panels bar = 50 µm, lower panels bar = 20 µm. (**b**) Pericyte coverage of capillaries in the second capillary layer; n = five animals per group, with three to four images evaluated per retina. (**c**) TNTs per field of view (fov); n = five animals per group, with three to four images evaluated per retina (Student’s *t* test). (**d**) The average length and the length distribution of TNTs from non-diabetic and Ins2^Akita^ mice; n = five animals per group, with three to four images evaluated per retina (Student’s *t* test), ** *p* < 0.01, **** *p* < 0.0001. (**e**) Morphology of pericytes (NG2+) detected in retinas from diabetic mice: (1) non-activated, endothelial cell-associated pericytes, (2) endothelial cell-associated pericytes that make TNTs, and (3) migrating pericytes.

**Figure 2 cells-13-01429-f002:**
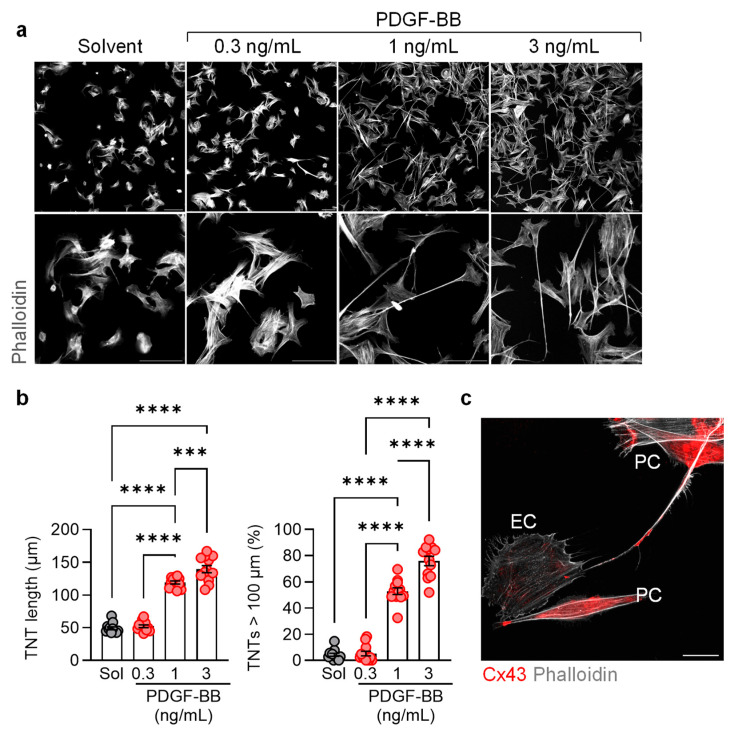
PDGF-BB-dependent in vitro pericyte tunneling. (**a**) Tunneling nanotubes (TNTs) induced by PDGF-BB (0.3–3 ng/mL, 6 h); bar = 50 µm. Comparable results were obtained using five additional cell batches. (**b**) Average TNT length per field of view (fov) and percentage of TNT’s longer than 100 µm; n = six independent cell batches with two evaluated images per batch (one-way ANOVA and Tukey’s multiple comparisons test); *** *p* < 0.001, **** *p* < 0.0001. (**c**) Connexin 43 (Cx43) in pericytes (PC), pericyte-derived TNTs, and an endothelial cell (EC). Comparable images were obtained using an additional three cell batches; bar = 20 µm.

**Figure 3 cells-13-01429-f003:**
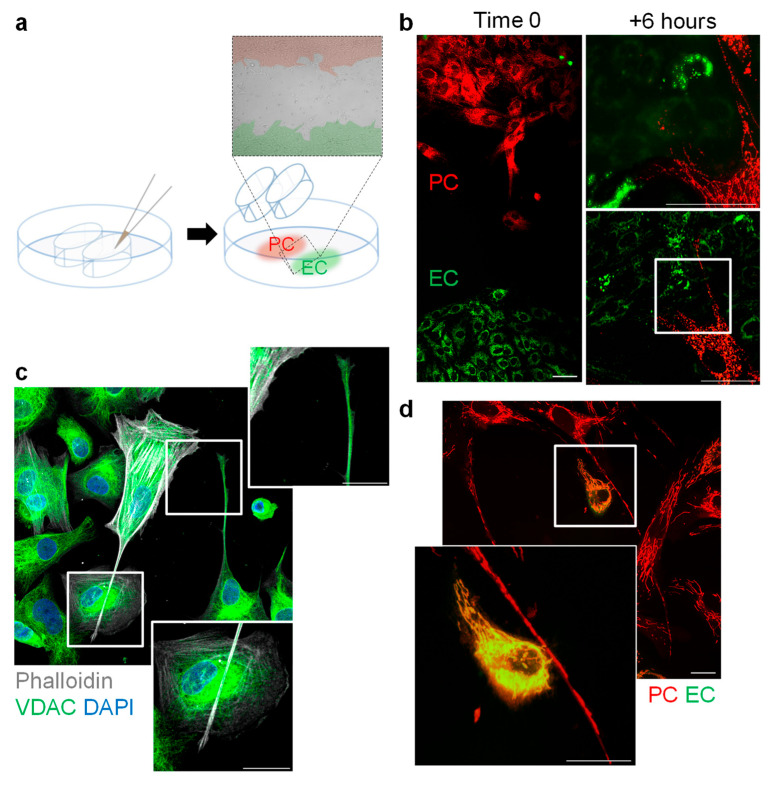
Transfer of mitochondria from pericytes to endothelial cells in vitro. (**a**) Scheme showing the migration chamber used to study the early contact of MitoTracker Red-labelled pericytes (PC) with MitoTracker Green-loaded endothelial cells (EC) for up to 6 h. (**b**) Labelled mitochondria in pericyte TNTs during first contact with endothelial cells in the presence of PDGF-BB (1 ng/mL) for six hours. Comparable results were made using five independent cell batches; bar = 20 µm. (**c**) Presence of the VDAC in pericyte TNTs. Comparable observations were made using four additional cell batches; bar = 20 µm. (**d**) Transfer of pericyte-derived mitochondria (red) to endothelial cells (green) in cells cocultured for 6 h in the presence of PDGF-BB (1 ng/mL); bar = 50 µm in overview, 10 µm in magnification. Comparable observations were made using six additional cell batches.

**Figure 4 cells-13-01429-f004:**
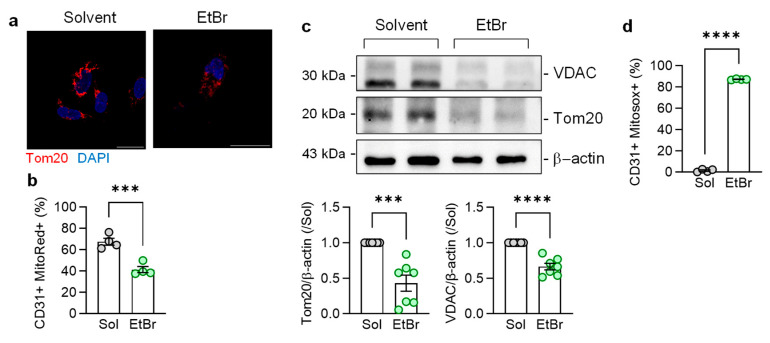
Consequences of EtBr-induced endothelial cell mitophagy on mitochondrial network integrity and metabolism. (**a**) Mitochondrial network in endothelial cells cultured in the presence of EtBr for 14 days, as visualized using Tom20. The images were taken 6 h after EtBr washout; bar = 20 µm. Comparable results were obtained in five additional cell batches. (**b**) FACs-based quantification of mitrotracker red (MitoRed) in CD31+ endothelial cells cultured in the absence and presence of EtBr; n = four independent cell batches (Student’s *t* test). (**c**) Expression of VDAC1 and Tom20 in endothelial cells cultured in the presence of solvent or EtBr for 14 days; n = seven independent cell batches (Student’s *t* test). (**d**) FACs-based quantification of mitochondrial reactive oxygen species (MitoSox) in CD31+ endothelial cells cultured in the absence and presence of EtBr; n = four independent cell batches (Student’s *t* test). *** *p* < 0.001, **** *p* < 0.0001.

**Figure 5 cells-13-01429-f005:**
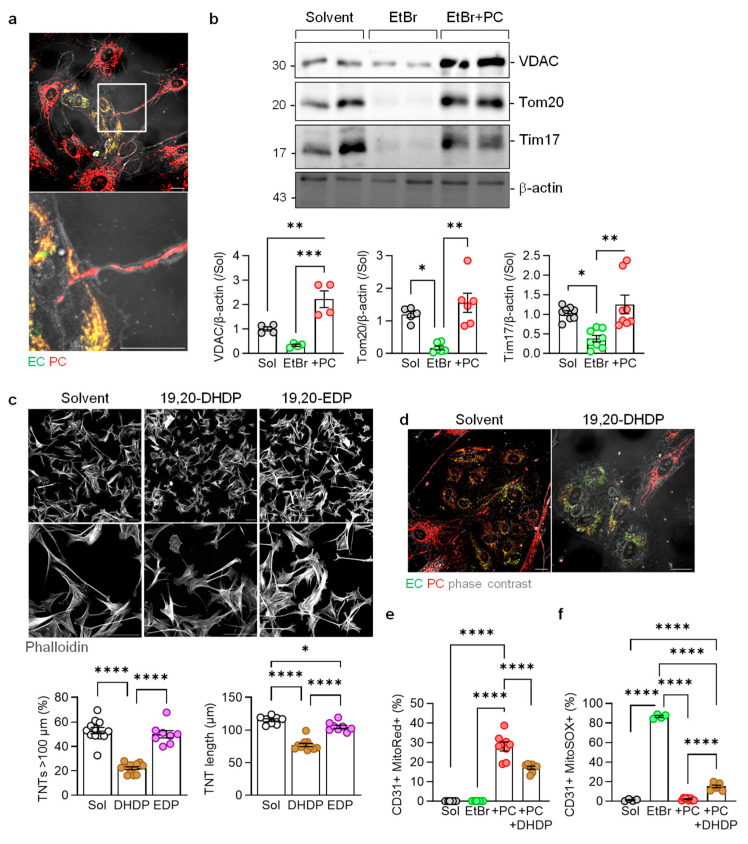
Transfer of mitochondria between pericytes and endothelial cells. (**a**) Mitochondrial transfer between pericytes (red) and EtBr-treated endothelial cells (green). The phase contrast images show mitochondria-filled TNTs emanating from pericytes towards endothelial cells. Comparable results were obtained in three additional experiments using independent cell batches; bar = 20 µm. (**b**) Mitochondrial proteins in endothelial cells treated with the solvent or EtBr or treated with EtBr prior to washout and culture with pericytes for 6 h (+PC); n = four independent cell batches (one-way ANOVA and Tukey’s multiple comparisons test). (**c**) TNT formation by PDGF-treated pericytes cultured in the presence of the solvent (Sol, 0.1% DMSO), 19,20-EDP, or 19,20-DHDP (10 µmol/L, 6 h). Experiments were performed in the presence of an sEH inhibitor; n = four independent cell batches with two images evaluated per cell batch (one-way ANOVA and Tukey’s multiple comparisons test). (**d**) Mitochondrial transfer from pericytes (red) to endothelial cells (green) in the absence and presence of 19,20-DHDP (10 µmol/L). (**e**,**f**) Transfer of mitochondria from Mitotracker Red (MitoRed)-labelled pericytes to CD31+ endothelial cells (**e**), and mitochondrial ROS production detected using Mitosox in CD31+ cells (**f**); n = four to eight independent cell batches (one-way ANOVA and Tukey’s multiple comparisons test). * *p* < 0.05, ** *p* < 0.01, *** *p* < 0.001, **** *p* < 0.0001.

**Figure 6 cells-13-01429-f006:**
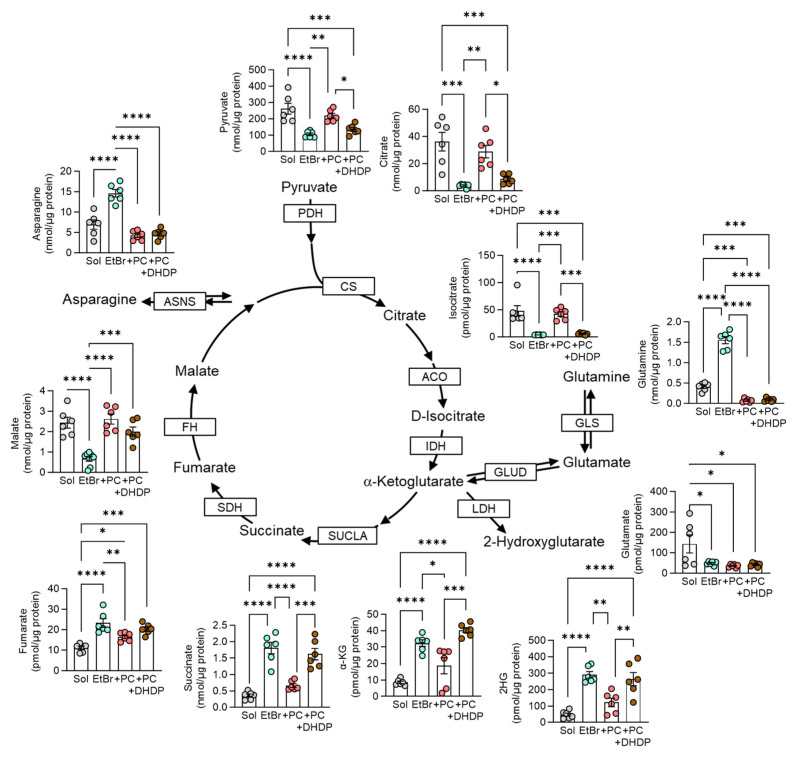
Rescue of endothelial cell metabolism by pericyte-derived mitochondria. Mitochondrial intact (Sol) or mitochondria-depleted (EtBr) endothelial cells were cocultured with pericytes (+PC) in the absence and presence of 19,20-DHDP for 36 h. Intermediates of the TCA cycle, asparagine, and glutamine metabolism were measured by LC-MS/MS; n = six independent cell batches (one-way ANOVA and Tukey’s multiple comparisons test). * *p* < 0.05, ** *p* < 0.01, *** *p* < 0.001, **** *p* < 0.0001. ACO: aconitase, ASNS: asparagine synthetase, CS: citrate synthase, FH: fumarate hydratase, GLS: glutaminase, GLUD: glutamate dehydrogenase, IDH: isocitrate dehydrogenase, LDH: lactate dehydrogenase, PDH: pyruvate dehydrogenase, SDH: succinate dehydrogenase, SUCLA: succinate-CoA ligase.

**Figure 7 cells-13-01429-f007:**
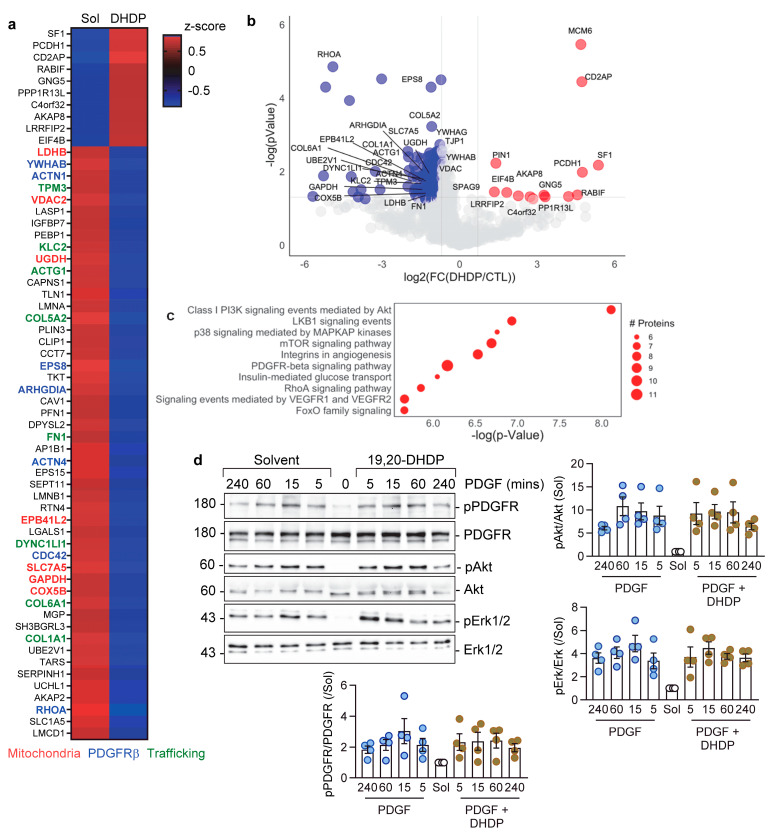
Effect of 19,20-DHDP on TNT formation and proteins in endothelial cell–pericyte junctions. (**a**) Heatmap showing the 50 pericyte–endothelial cell junctions most affected by 19,20-DHDP (10 µM, 16 h). Mitochondrial proteins are highlighted in red, PDGFR-β signaling in blue, and trafficking-related proteins in green; n = three independent cell batches. (**b**) Volcano plot showing endothelial cell–pericyte junctional proteins in co-cultures of endothelial cells and pericytes, comparing 19,20-DHDP with non-treated fractions (19,20-DHDP/Solvent). (**c**) Pathway enrichment analysis (NCI-Nature) of proteins significantly decreased in the junctional space after exposure to 19,20-DHDP. (**d**) Representative Western blot showing the effect of PDGFBB (1 ng/mL, for up to 240 min) on the phosphorylation of the PDGF receptor, Akt, and Erk1/2 in pericytes in the absence and presence of 19,20-DHDP; n = four independent cell batches.

## Data Availability

The datasets generated and/or analyzed during this study are either available from an open source or available from the corresponding author upon reasonable request.

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
