# Peer review of "Pericyte-to-Endothelial Cell Communication via Tunneling Nanotubes Is Disrupted by a Diol of Docosahexaenoic Acid"

_cells, 2024, doi:10.3390/cells13171429_

Round 1

Reviewer 1 Report

Comments and Suggestions for Authors

This is an interesting and innovative paper on the interaction between pericytes and endothelial cells, especially the role of a natural diols derived from DHA. Overall, the experiments seem well done and the results support the conclusions. Thus, it should be published. However, I have few comments/questions.

1. A lot of the results from exposure to the diols are compared to the solvent. It seems to me very far conditions (solvent vs diol) to be compared. I will prefer to see comparison with the fatty acid (DHA) or another DHA diol that do not influence the TNT. Looking back at the authors original publication (#19), they only explored the role of the 19,20-DHDP on pericytes. It seems very restrictive especially that other diols form DHA are produced faster by the sEH.

2. In the "protein experiment" (figure 7), the authors isolated TNTs and analyzed the protein content. How the authors confirm that the TNT samples were not contaminated by cytosolic proteins (or membrane proteins) from either cell types? Probably providing a analysis of the cytosol of both cell types could show the specificity of protein content of the TNTs. 

Comments on the Quality of English Language

Overall, the English is very good. I found few typos that could be easily corrected. For example on line 461 it is missing the "o" of formation or on line 516, "inasmuch" missing spaces between words. There are probably few more typos. 

Author Response

  1. A lot of the results from exposure to the diols are compared to the solvent. It seems to me very far conditions (solvent vs diol) to be compared. I will prefer to see comparison with the fatty acid (DHA) or another DHA diol that do not influence the TNT. Looking back at the authors original publication (#19), they only explored the role of the 19,20-DHDP on pericytes. It seems very restrictive especially that other diols form DHA are produced faster by the sEH.

Reply: As we outlined in the text, the reason for focusing on 19,20-DHDP was because of its role in the retina and diabetic retinopathy. 19,20-DHDP was the only significantly altered PUFA diol in non-proliferative retinopathy in mouse and man. We did perform initial experiments with the sEH substrate 19,20-EDP as well. This data was initially not shown but has now been included in the revised version of Figure 5.

  1. In the "protein experiment" (figure 7), the authors isolated TNTs and analyzed the protein content. How the authors confirm that the TNT samples were not contaminated by cytosolic proteins (or membrane proteins) from either cell types? Probably providing a analysis of the cytosol of both cell types could show the specificity of protein content of the TNTs. 

Reply: Figure 7 showed endothelial cell-pericyte contacts that we generated using a filter system with endothelial cells on one side and pericytes on the other. Cells were scrapped and proteins in the filter pores were isolated (please see page 3/20). This method really only enriches for junctional proteins i.e. proteins in the membrane and the contents of endothelial cell pericyte junctions, and as we realise this we did not claim that we looked at TNTs per se. We have no methods for isolating pure TNTs.

Comments on the Quality of English Language

Overall, the English is very good. I found few typos that could be easily corrected. For example on line 461 it is missing the "o" of formation or on line 516, "inasmuch" missing spaces between words. There are probably few more typos.

Reply: We hope that we have found all of the typos: We suspect the issues most probably related to our English spelling e.g. signalling, but “inasmuch” is correct as used. We have corrected the spelling to US English throughout.

Reviewer 2 Report

Comments and Suggestions for Authors

The manuscript by Kempf et al investigated the formation and function of pericyte tunneling nanotubes and their role in regulating endothelial cell metabolism, as well as the effect of 19,20DHDP on pericyteendothelial cell junctions. They found that pericytes formed TNTs facilitated mitochondrial transport from pericytes to endothelial cells and restored mitochondrial network and metabolism in mitochondriadepleted endothelial cells. Whereas, 19,20DHDP treatment prevented TNT formation and metabolic rescue in mitochondriadepleted endothelial cells, along with changes in the protein composition of pericyteendothelial cell junctions. This is an interesting and well-performed study. The reviewer has the following comments.

1)    In the Methods section 2.3, it stated that the experiments were performed in the presence of sEH inhibitor. However, no such data was presented.

2)    Mitochondrial depletion by EtBr treatment seems to have significantly affected the expression of b-actin, which makes it unreliable as a loading control. It is better to use other housekeeping proteins as loading control.

3)    In Figure 7d, in the presence of 19,20DHDP, PDGF treatment seems to induce a more significant phosphorylation of ERK1/2 at 5 min and 15 min. Please provide quantification data for this figure. If it is true, please revise the result and discuss the different responses in the phosphorylation of Akt and ERK after treatment.  

Comments on the Quality of English Language

Page 2, line 83, "pericytes where used until passage seven", please check spelling.

Author Response

  • In the Methods section 2.3, it stated that the experiments were performed in the presence of sEH inhibitor. However, no such data was presented.

Reply: We performed initial experiments with the sEH substrate; 19,20-EDP, as well as 19,20-DHDP and to ensure there was no conversion of the epoxide to the diol, experiments were performed in the presence of a sEH inhibitor. There was no impact of the sEH inhibitor per se on TNT formation by pericytes in vitro. Data showing the lack of impact of 19,20-EDP on TNT formation has been included in the revised version of Figure 5.

  • Mitochondrial depletion by EtBr treatment seems to have significantly affected the expression of b-actin, which makes it unreliable as a loading control. It is better to use other housekeeping proteins as loading control.

Reply: We appreciate the reviewers point and have replaced the b-actin blot shown as requested.

3)    In Figure 7d, in the presence of 19,20‐DHDP, PDGF treatment seems to induce a more significant phosphorylation of ERK1/2 at 5 min and 15 min. Please provide quantification data for this figure. If it is true, please revise the result and discuss the different responses in the phosphorylation of Akt and ERK after treatment.

Reply: The reviewers impression was incorrect as there was no significant difference in the responses with and without 19,20-DHDP. We have included the quantification in the revised version of Figure 7.

Comments on the Quality of English Language

Page 2, line 83, "pericytes where used until passage seven", please check spelling

Reply: We hope that we have found all of the typos but in that case the spelling was correct.

Round 2

Reviewer 2 Report

Comments and Suggestions for Authors

No further comments.